# An open source automated tumor infiltrating lymphocyte algorithm for prognosis in melanoma

Balazs Acs [1,2], Fahad Shabbir Ahmed [1], Swati Gupta[1], Pok Fai Wong[1], Robyn D. Gartrell[3],
Jaya Sarin Pradhan[4], Emanuelle M. Rizk[5], Bonnie Gould Rothberg[6], Yvonne M. Saenger[5] & David L. Rimm [1,6]*

Assessment of tumor infiltrating lymphocytes (TILs) as a prognostic variable in melanoma has not seen broad adoption due to lack of standardization. Automation could represent a solution. Here, using open source software, we build an algorithm for image-based automated assessment of TILs on hematoxylin-eosin stained sections in melanoma. Using a retrospective collection of 641 melanoma patients comprising four independent cohorts; one training set (N = 227) and three validation cohorts (N = 137, N = 201, N = 76) from 2 institutions, we show that the automated TIL scoring algorithm separates patients into favorable and poor prognosis cohorts, where higher TILs scores were associated with favorable prognosis. In multivariable analyses, automated TIL scores show an independent association with disease-specific overall survival. Therefore, the open source, automated TIL scoring is an independent prognostic marker in melanoma. With further study, we believe that this algorithm could be useful to define a subset of patients that could potentially be spared immunotherapy.

[1] Department of Pathology, Yale School of Medicine, New Haven, CT 06510, USA. [2] Department of Oncology and Pathology, Karolinska Institutet, Stockholm, Sweden. [3] Department of Medicine, Division of Hematology/Oncology, Columbia University Medical Center/New York Presbyterian, New York, NY, USA. [4] Department of Pathology and Cell Biology, Division of Oral and Maxillofacial Pathology, Columbia University Irving Medical Center/New York Presbyterian, New York, NY, USA. [5] Department of Medicine, Division of Hematology/Oncology, Columbia University Irving Medical Center/New York Presbyterian, New York, NY, USA. [6] Department of Medicine, Yale School of Medicine, New Haven, CT 06510, USA. *email: david.rimm@yale.edu

The development and progression of malignant tumors requires interaction with other cells in the tumor micro-environment, including immune cells[1]. Due to altered protein expression by tumor cells, the immune system can recognize malignancy and induce an immune response[2,3]. Growing evidence from several studies supports that assessment of tumor-infiltrating lymphocytes (TIL) has prognostic significance in many tumor types[4]. Recent studies have shown that immunotherapy in the adjuvant setting benefits only one in five patients[5,6]. These studies show significant toxicities including one in five with significant hypothyroidism and one in hundred fatalities[5,6] These observations warrant the introduction of a prognostic test that can determine which patients can be spared treatment.

TILs have traditionally been scored semi-quantitatively on standard hematoxylin and eosin (H&E)-stained section as absent, non-brisk, or brisk[7]. Some departments use a four-tier grading system that includes both TIL distribution and density[8]. These scoring methods have been established for many years, but have not seen broad adoption in clinical decision-making due to lack of standardization between institutions and concerns regarding reproducibility between pathologists[9]. In breast cancer, the International TIL working group has put serious efforts on the standardization of TILs scoring and published a guideline that is made for visual evaluation of HE sections by pathologists[10,11]. Although the robustness of the TIL guideline has been shown in international ring trials[12], it is unlikely that a subjective method will be sufficiently accurate and reproducible to be used to select patients to be spared from therapy[13]. Digital-image analysis (DIA) may present a solution to this problem. DIA can facilitate the analysis of complex spatial patterns, and could provide standardized metrics for rigorous validation[13]. Classical segmentation and neural networks approaches have been applied widely in various DIA platforms to overcome cell classification challenges[14,15]. DIA platforms are able to evaluate TILs, but no study has published yet to show robustness of automated TIL scoring in melanoma. Therefore, we built an algorithm (called eTIL%) for image-based, automated assessment of tumor-infiltrating lymphocytes (TILs) on H&E-stained sections in melanoma.

## Results

### Performance of automated TIL scoring in TMA cohorts.
The best threshold with statistical significance after cross-validation in Xtile occurred at 16.6% ($p = 0.01$). The automated TIL scoring algorithm showed high eTIL% patients showed statistically significantly better disease-specific overall survival (DSOS) compared with low eTIL% (Log rank $p = 0.007$; HR = 0.420, CI = 0.220–0.802; Fig. 1a). Higher TILs scores were associated with favorable prognosis. In contrast, traditional pathologist-read visual assessment of TILs, modified to be assessed on a TMA, failed to distinguish patient cohorts with different DSOS (Log rank $p = 0.821$; HR = 0.871, CI = 0.461–1.646; Fig. 1b). This same cohort was also tested for expression of CD4, CD8, and CD20 using quantitative fluorescent methods[16]. These results show a weak correlation of each of these lymphocytic subtypes to eTIL% (Fig. 1c–e), but none were prognostic, although CD4 trended toward, but did not reach significance. We found a significant relationship and fair correlation between eTILs and the sum of CD4 and CD8 (Spearman $r = 0.466$, $p < 0.001$). Using the 16.6% cut point defined in cohort #1 on TMA images from cohorts #2 and #3, eTIL% separated patients into favorable and unfavorable prognostic subsets (cohort #2: Log rank $p < 0.001$; HR = 0.397, CI = 0.242–0.651; Fig. 2a; cohort #3: Log rank $p = 0.002$; HR = 0.409, CI = 0.226–0.741; Fig. 2b). The

clinicopathological factors were prognostic in univariate analysis (Supplementary Table 1). We also investigated the association between eTIL% and the clinicopathological factors. eTIL% score was higher in cases with absent ulceration and smaller tumor depth in cohorts #1, #2, and #3. No significant association was found between eTIL% and any clinicopathological factor in cohort #4 (Supplementary Fig. 1).

### Performance of automated TIL scoring in whole-slide cohorts.
In the clinical setting, assessments must be done on whole-tissue sections. So, we applied the NN192 algorithm to every field of view on whole-slide images from cohorts #2 and #4. Again eTIL% high, above the 16.6%, showed significantly favorable DSOS in both cohorts (cohort #2: Log rank $p < 0.001$; HR = 0.119, CI = 0.057–0.245, Fig. 2c) and cohort #4 (Log rank $p = 0.036$; HR = 0.391, CI = 0.157–0.974; Fig. 2d). The eTIL% score remained significant in both cohorts even after adjusting for age, sex, ulceration, stage (using the system concurrent with tissue collection dates), Breslow depth, Clarke levels, and location of primary tumor (Table 1). We found significant difference and low reproducibility between the corresponding TMA and WSI cases regarding eTIL% scores (Supplementary Fig. 2).

## Discussion

Although it has long been acknowledged that TILs might provide prognostic and predictive information in melanoma[11,17–19], it has not been widely adopted for clinical melanoma management due to inter-operator and inter-institutional variability[20]. Key contributors to variability include both preanalytical and analytical steps, and most significantly, lack of reproducibility in scoring[13,21]. To address the issue of subjective variance, studies have been published proposing TIL quantification using convolutional networks on image-based immunohistochemistry (IHC)-stained sections in gastric, breast, prostate, and colon cancer[22–24]. However, in these investigations, the TIL quantification relies on the detection of IHC markers, such as CD3 and CD8 and in some cases, depends on their localization with respect to the leading edge of the tumor[25]. In our previous work, we showed that digital-image analysis of CD8 detection is associated with anti-PD-1 response in metastatic melanoma, but is not prognostic[16]. Here, the prognostic value of the eTIL% may arise from the combination of a range of lymphocytes combined with the normalization by the number of adjacent melanocytes. While subtyping of TILs may become important in immunotherapy prediction, also requiring image analysis, a combination of eTIL% and quantitative CD8 might be the best approach for finding patients that can be spared immunotherapy in the adjuvant setting. In our study, we found a significant relationship and weak–fair correlation between eTILs and CD4, CD8 and CD20 expression. A possible explanation could be immune heterogeneity in the tumor as the HE and CD4, CD8, CD20 IF stainings were not performed on serial sections. Another possible explanation is that CD4, CD8, and CD20 (immune markers) scores were calculated differently during the automated quantitative fluorescent investigation. Regarding immune markers, scores were calculated in terms of the number of cells positive for the marker of interest as a percentage of the cell population in which it was measured, while eTIL% was defined as (TILs/TILs + tumor cells) × 100.

Detection of TILs might provide a cost-effective and robust prognostic marker, especially when no additional molecular tests are available. In a recent study by Heindl et al., the prognostic potential of automated TIL scoring was investigated in ER-positive (+) breast cancer[26]. The authors used cell segmentation–classification approach for DIA on H&E-stained sections, and they found that immune spatial clustering scores

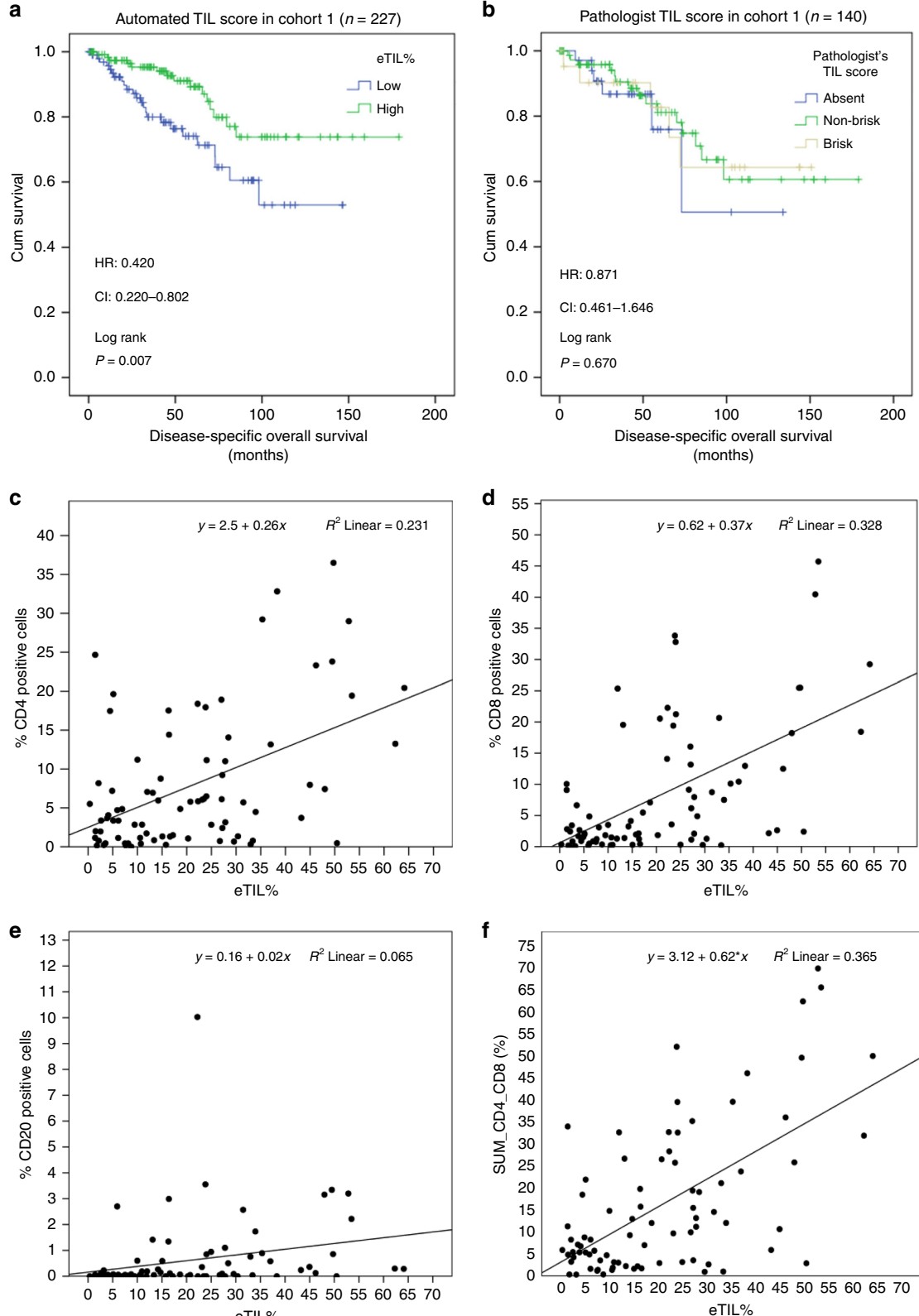

**Fig. 1** Tumor-infiltrating lymphocytes (TIL) scores in cohort #1. The prognostic potential of automated TIL scores (**a**) and pathologist's TIL scores (**b**) in cohort #1. Correlation between eTIL% scores and CD4 positive (+), CD8+, and CD20+ immune cells measured by quantitative immunofluorescence (**c-f**).

obtained by DIA were linked with poor recurrence-free survival after endocrine therapy in ER+ breast cancer[26]. In our study, we used a similar approach, unsupervised nuclei segmentation followed by neural network machine-learning-based cell

classification. The advantage of segmentation-based object detection is that it requires relatively smaller of training sets[13,27,28]. On the other hand, segmentation sensitivity and classification accuracy are subject to biological and technical

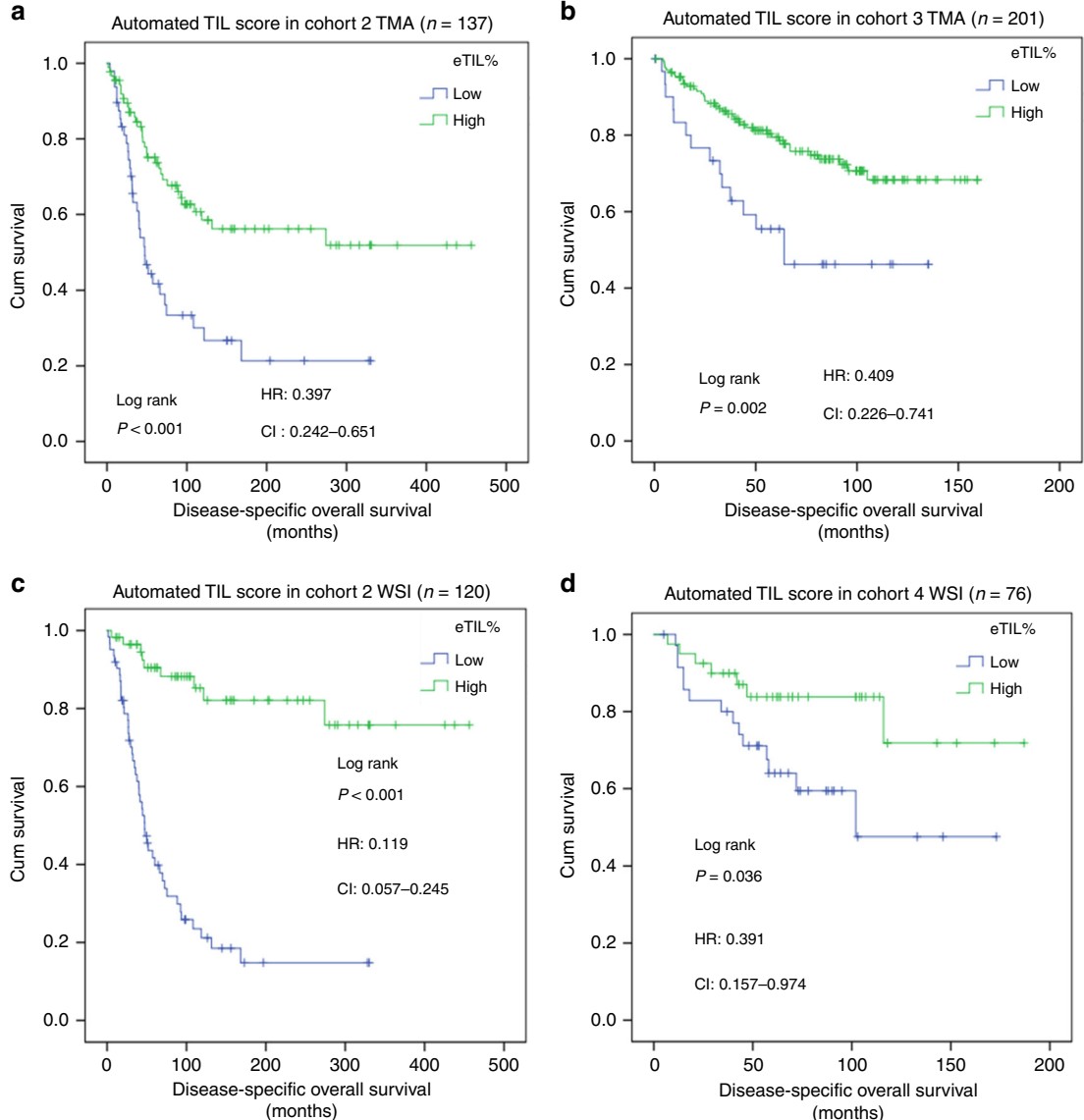

**Fig. 2** Validation of automated tumor-infiltrating lymphocytes (TIL) algorithm in three independent cohorts. The prognostic potential of automated TIL scores performed on tissue microarray (TMA slides) (**a**, **b**) and whole slides (**c**, **d**) in cohorts #2, #3, and #4.

**Table 1 Multivariate Cox-regression analysis of eTIL% score and the clinicopathological factors in whole-slide cohorts #2 and #4 regarding disease-specific overall survival.**

| Prognostic factor | Cohort #2 (n = 120) | | | Cohort #4 (n = 76) | | |
|---|---|---|---|---|---|---|
| | HR | 95% CI | *p*-value | HR | 95% CI | *p*-value |
| Age | 0.985 | 0.958–1.013 | 0.289 | 1.014 | 0.979–1.049 | 0.446 |
| Sex | 1.601 | 0.876–2.927 | 0.126 | 0.497 | 0.129–1.914 | 0.310 |
| Tumor depth | 1.156 | 0.747–1.788 | 0.516 | 1.441 | 0.514–4.043 | 0.488 |
| Clarke levels | 1.369 | 0.708–2.645 | 0.350 | NA | NA | NA |
| Ulceration | 1.190 | 0.623–2.272 | 0.599 | 1.397 | 0.454–4.293 | 0.560 |
| Stage | 1.389 | 0.990–1.948 | 0.057 | 4.463 | 1.365–14.600 | 0.013 |
| Location of primary tumor | NA | NA | NA | 0.588 | 0.226–1.529 | 0.276 |
| Automated TIL score | 0.143 | 0.063–0.326 | <0.001 | 0.326 | 0.122–0.874 | 0.026 |

image variation, that might result in overfitting for the training set[13,27,28]. For this reason, we tested the performance of our algorithm in three, independent populations that vary in time of diagnosis, tissue preparation, and H&E staining investigated on both TMA and whole-slide sections. We found significant prognostic potential of automated TIL scores in all the validation sets.

There are a number of limitations to this work. Most significantly, the cohorts were all retrospectively collected, including

**Table 2 Clinicopathological data of the patients.**

| Total n = 641 | | | Cohort 1 | | Cohort 2 | | Cohort 3 | | Cohort 4 | |
|---|---|---|---|---|---|---|---|---|---|---|
| Patients | | n, % | 227 | 100% | 137 | 100% | 201 | 100% | 76 | 100% |
| Age | Mean ± SD, range | | 64 ± 16.5 | 18–97 | 59 ± 14.6 | 25–87 | 59 ± 18.1 | 19–88 | 65 ± 15.2 | 22–96 |
| Sex | Male | n, % | 136 | 59.9% | 70 | 51.1% | 117 | 58.2% | 59 | 77.6% |
| | Female | n, % | 91 | 40.1% | 67 | 48.9% | 84 | 41.8% | 17 | 22.4% |
| Tumor depth | Unknown | n, % | 0 | 0% | 6 | 4.4% | 13 | 6.5% | 0 | 0% |
| | ≥2 mm | n, % | 126 | 55.5% | 79 | 57.7% | 58 | 28.9% | 51 | 67.1% |
| | <2 mm | n, % | 101 | 44.5% | 52 | 37.9% | 130 | 64.6% | 25 | 32.9% |
| Clarke levels | Unknown | n, % | 42 | 18.5% | 6 | 4.4% | 19 | 9.5% | NA | NA |
| | I | n, % | 1 | 0.4% | 0 | 0% | 1 | 0.5 | NA | NA |
| | II | n, % | 24 | 10.6% | 17 | 12.4% | 2 | 1% | NA | NA |
| | III | n, % | 25 | 11% | 46 | 33.6% | 30 | 14.9% | NA | NA |
| | IV | n, % | 113 | 49.8% | 47 | 34.3% | 145 | 72.1% | NA | NA |
| | V | n, % | 22 | 9.7% | 21 | 15.3% | 4 | 2% | NA | NA |
| Ulceration | Unknown | n, % | 88 | 38.8% | 8 | 5.8% | 13 | 6.5% | 5 | 6.6% |
| | Yes | n, % | 65 | 28.6% | 51 | 37.3% | 37 | 18.4% | 41 | 53.9% |
| | No | n, % | 74 | 32.6% | 78 | 56.9% | 151 | 75.1% | 30 | 39.5% |
| Stage | Unknown | n, % | 5 | 2.2% | 15 | 11% | NA | NA | 2 | 2.6% |
| | I | n, % | 85 | 37.5% | 98 | 71.5% | NA | NA | 0 | 0% |
| | II | n, % | 93 | 41.0% | 6 | 4.4% | NA | NA | 59 | 77.6% |
| | III | n, % | 42 | 18.5% | 15 | 11% | NA | NA | 15 | 19.8% |
| | IV | n, % | 2 | 0.8% | 3 | 2.1% | NA | NA | 0 | 0% |
| Location of primary tumor | Unknown | n, % | NA | NA | NA | NA | 13 | 6.5% | 2 | 2.6% |
| | Trunk | n, % | NA | NA | NA | NA | 102 | 50.7% | 41 | 53.9% |
| | Extremity | n, % | NA | NA | NA | NA | 86 | 42.8% | 33 | 43.5% |
| Follow-up (months) | DSOS | Median, IQT | 44 | 49.3 | 59.9 | 97.7 | 79 | 60.4 | 61.5 | 55 |

*DSOS* disease-specific overall survival, *IQT* interquartile range.

the oldest tumors dating back to the 1990s. While the treatments pre-dating immune therapy were approximately equivalent with respect to outcome, none of the cohorts are uniformly treated as would be seen in clinical trials. Another limitation is that all cases were scanned using a single scanner manufacturer, albeit at different institutions and at different magnifications. While the software algorithm NN192 appears to perform equivalently, it is not clear if the algorithm would perform similarly with images from other scanners. Further studies are needed to validate the prognostic potential of this algorithm in more independent cohorts and to determine the technical requirements for image acquisition. The performance of machine-learning-based classification depends on the training[13]. Even though this is an open-source software, and thus broadly available, quality control and systematic performance evaluations must be implemented prior to the use of eTIL% scoring in adjuvant immunotherapy cohorts or in clinical practice. In our study, a pathologist performed quality control of the three algorithms to classify detected cells. Although we are not attempting to validate TMA technology here, it is notable that we found only modest correlation among TMA and whole-slide cases eTIL% scores. This might be due to the fact that the analyzed tumor area on an average melanoma whole slide is 20–25-fold larger than a TMA spot. However, eTIL % was prognostic irrespective of scoring on TMAs or whole-slide cases.

In conclusion, this study shows that an automated TIL score is a robust, independent prognostic marker in melanoma. With validation, we believe that this approach could be tested in the immunotherapy adjuvant setting to define a subset of patients that could potentially be spared treatment and its significant toxicities.

## Methods

**Patient cohorts and tissue preparation**. Our retrospective collection of 641 melanoma tumors included four independent cohorts, three from Yale New Haven Hospital (YNHH), and one from Columbia University Irving Medical Center (CUMC). The training set (cohort #1) consists of 227 patients diagnosed between

1993 and 2005 with 44 months median follow-up. The validation sets include cohort #2: 137 patients diagnosed between 1999 and 2011 with 59.9 months median follow-up; cohort #3: 201 patients diagnosed between 1981 and 2010 with 79 months median follow-up; and cohort #4: 76 patients from CUMC, diagnosed between 2000 and 2012 with 61.5 months median follow-up (Table 2). Cohorts #1, #2, and #3 were assessed as tissue microarrays (TMAs). Representative tumor areas were selected by pathologists based on H&E-stained slides. Duplicate cores (each 0.6 mm in diameter) were punched from each case. The H&E-stained sections of the TMAs were scanned for analysis in this study. For cohorts #2 and #4, every field from the original H&E slides were assessed resulting in average areas of assessment of 7.46 mm$^2$ and 5.88 mm$^2$ compared with 0.28 mm$^2$ for TMA spots (Supplementary Fig. 3). One whole slide per patient selected by a pathologist was selected for the study. This study have complied with all relevant ethical regulations, and it was approved by the Yale Human Investigation Committee under protocol #9505008219 and Columbia IRB protocol #AAO2758. Patients in each cohort provided informed consent or (especially for older tissues) the tissue was obtained through Yale Human Investigation Committee protocol #9505008219 which allows waiver of consent in some cases. Specifically, waiver of consent was in place for all patients at the time the Yale cohort #1 collected in 2005. Older tissues in Yale Cohorts #1 and #3 were also collected with waiver of consent and for all deceased patients in all Yale cohorts. For the criterion standard, a pathologist scored TILs in cohort #1 as absent, non-brisk, or brisk. The reporting recommendations for tumor marker prognostic studies (REMARK) were followed in our study[29].

**Digital-image analysis (DIA)**. In cohorts #1, #2, and #3, the Aperio ScanScope XT platform (Leica Biosystems, Wetzlar, Germany) was used at ×20 to digitize the slides with a pixel size of 0.4986 μm × 0.4986 μm. In cohort #4, Aperio ScanScope XT platform (Leica Biosystems, Wetzlar, Germany) was used at ×40 to digitize the slides with a pixel size of 0.2500 μm × 0.2500 μm. The QuPath open-source software platform[30] was used to build automated TIL scoring algorithm. As the date of H&E staining varied both between and within cohorts, we refined the H&E stain estimates for each digitized slide (using the "estimate stain vectors" command in QuPath). We used watershed cell detection[31] to segment the cells in the image with the following settings: Detection image: hematoxylin OD; requested pixel size: 0.5 μm; background radius: 8 μm; median filter radius: 0 μm; sigma: 1.5 μm; minimum cell area: 10 μm$^2$; maximum cell area: 400 μm$^2$; threshold: 0.1; maximum background intensity: 2. The quality control of the cell segmentation was performed by a pathologist. In order to classify detected cells into tumor cells, immune cells (TILs), stromal cells, and others (false detections, background) (Fig. 3), we used neural network[32] as a machine-learning method with eight hidden layers (maximum iterations: 100). The features used in the classification are described in Supplementary Table 2. In order to help the algorithm perform an accurate classification, we also added smoothed object features at 25 μm and 50 μm radius to supplement the existing measurements of individual cells. A number of rounds of

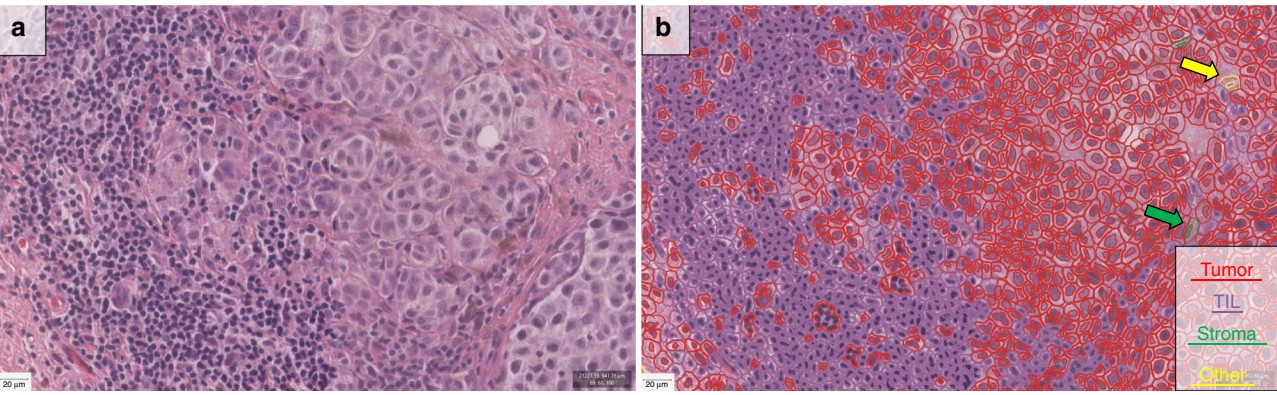

**Fig. 3** Representative picture of a sample melanoma case showing the H&E image (Zoom: ×20, **a**) and the digital-image analysis (DIA) mask (**b**). Scale bar represents 20 μm. Using the NN192 algorithm, segmentation shows red indicates tumor cells, purple marks immune cells, green corresponds to stromal cells, and yellow indicate others (false cell detections or unknown or background). Since stromal and "other" cells are rare, large arrows are included to show example cells.

optimization were required to achieve best results on the training set culminating in an algorithm called "NN192" that calculates the percentage of machine defined TILs calculated as follows: (TILs/TILs + Tumor cells) × 100 called "eTIL%". The quality control of the algorithm to classify detected cells was performed by a pathologist. In the whole-slide cohorts (cohorts #2 and #4), the analysis was run on the entire tumor as defined based on the pathologists' markings (Supplementary Fig. 3).

**Statistical analysis**. For statistical analysis, SPSS 22 software (IBM, Armonk, USA) software was used. The statistically significant cutoff for TILs scores was determined with X-tile software that uses outcome information to define threshold[33]. $\chi^2$ value was calculated for every possible division of the population and best cutoff (highest $\chi^2$ value) went under cross-validation to assess statistical significance by using the cut point derived from a training set to parse a separate validation set. Kaplan–Meier analysis supported with Log-rank test was executed to assess prognostic potential. To test independent prognostic potential, multivariate Cox-regression analysis was applied. Disease-specific overall survival (DSOS) was defined as the elapsed time from the date of primary diagnosis of the tumor to the date of death caused by melanoma, or when patients were last censored if died of non-melanoma cause or still alive. Mann–Whitney test was used to investigate the association between TILs scores and clinicopathological factors. To test the reproducibility between the corresponding TMA and whole-slide cases regarding TILs, intraclass correlation was supported by scatterplot and Wilcoxon signed-rank test. In all statistical analysis, the level of significance was set at $p < 0.05$.

**Reporting summary**. Further information on research design is available in the Nature Research Reporting Summary linked to this article.

## Data availability
The data is within the Article and Supplementary Information files and available from the authors upon request but the data from Columbia may require data transfer agreements. No personalized health information will be shared.

## Code availability
Our TIL scoring algorithm for HE images of Melanoma has been deposited on GitHub: https://github.com/acsbal/Automated-TIL-scoring-QuPath-Classifier-for-Melanoma. The algorithm can be used in QuPath platform. The QuPath software may be downloaded at https://qupath.github.io/.

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

## Acknowledgements

Dr. B.A. was supported by the Fulbright Program and the Rosztoczy Foundation Scholarship Program. Dr. P.F.W. was supported by the Gruber Science Fellowship from the Gruber Foundation. This work was supported by Navigate BioPharma and grants from the NIH. Robyn Gartrell is supported by Swim Across America and the National Center for Advancing Translational Sciences, National Institutes of Health, through Grant Number KL2TR001874.

## Author contributions

Conception and design: B.A. and D.L.R. Case selection and clinical data collection: P.F.W., R.D.G., J.S.P., E.M.R., Y.M.S., BG.R. and D.L.R. Acquisition of digital images: B.A., F.A., and S.G. Setting up image analysis algorithm, training of the algorithms, TIL scoring: B.A. Statistical analysis: B.A. Drafting of the paper: B.A., D.L.R. Critical revision of the paper: all authors. Final approval of the paper: all authors. Study supervision: D.L.R.

## Competing interests

D.L.R. declares that he has served a consultant, advisor and/or servee on a Scientific Advisory Board for Amgen, Astra Zeneca, Agendia, Biocept, BMS, Cell Signaling Technology, Cepheid, Daiichi Sankyo, GSK, InVicro/Konica Minolta, Merck, NanoString, Perkin Elmer, PAIGE.AI, and Ultivue. He holds equity in PixelGear (start-up company related to direct tissue imaging) and Astra Zeneca, Cepheid, Navigate/Novartis, NextCure, Lilly, Ultivue, Ventana and Perkin Elmer/Akoya fund research in his lab. The remaining authors declare no competing interests.
