## [Peer Review File · Nature Communications]

Reviewers' comments:

Reviewer #1 (Remarks to the Author):

Thank you for addressing my comments. Overall, this was/is a well articulated manuscript and response letter. I think this study is novel and will be found to be of value to the community of pathologists and oncologists as it will indeed provide an inexpensive tool to tabulate TILs using routine H&E stained sections. I recommend acceptance. One important comment in response to your comments.

1. Originally, I commented, "There was poor correlation between CD3 density and the eTIL% score, which is surprising. Is there additional data available (CD45 for example) to more convincingly demonstrate that the eTILs% score is in fact reflective of the actual TILs?"

The authors response: "With apologies to the reviewer, we did not test CD3, only CD4, CD8 and CD20. Nor did we test CD45, but that would be missing the point. We are not trying to molecularly define TILs, but rather finding a morphologically defined criteria (algorithm) that appears to be better at predicting outcome than any molecular method we have tested."

With commensurate apologies to the authors, on page 8 of the manuscript, the authors state, "This same cohort was also tested for expression of CD3, CD4, CD8 and CD20 using quantitative fluorescent methods."

Only CD4, CD8, and CD20 are shown in Figure 2, but CD3 was/is stated to have been tested? In any case, the point was not that it would be important to molecularly define what eTILs% is tabulating per se (I agree that would be missing the point, but only sort of). The intention was to suggest that it would be important to confirm that eTILs% is actually a measure of 'immune cells'-not that it is important what the cells actually are, provided they are indeed 'immune cells'. The contention that eTILs% is a measure of L's is a bit shaky since eTILs% only weakly correlates with conventionally tabulated metrics of immune cells (CD4, CD8 in particular). Might eTILs% correlate with the sum of CD4 and CD8 (this should be comprise most all of the T cells and many but not all of the histiocytes/dendritic cells in the sample)?

Reviewer #3 (Remarks to the Author):

Review of An Open Source Automated Tumor Infiltrating Lymphocyte Algorithm for Prognosis

This manuscript has been revised and responses provided to the reviewer's comments. We respond as follows:

Clinical context

1. The authors have now clarified that based on this retrospective study eTIL% cannot yet be used as a decision aid to spare patients from adjuvant immunotherapy. This will require testing in a prospective clinical trial, but it is accepted that the current study provides useful baseline data for such a future study.

2. The additional details provided on development of the algorithm and features used for neural network classification are helpful (see below).

3. Whilst the authors state that eTIL% was prognostic irrespective of whether TMA or whole slide images were used, correlation between the two was not strong. This suggests there may be heterogeneity in eTIL% dependent on the region of tumour tested which could impact on the

prognostic scoring. Have the authors formally tested this using feature extraction at low and higher power? Future validation should determine the optimal number of tumour regions to test for most optimal utility of the tool.

Computational/Histopathology

4. Both the watershed and neural networks approaches have been applied widely in order contexts to overcome segmentation and classification challenges and there is little novelty in this regard. The authors should refer to these works in the introduction-(related works) section.

5. What is the reason that the authors used the aforementioned approaches and not any other approach (Graph Cuts, RCNN, FCN)?

6. From Figure 1B, it is clear shown that the watershed algorithm didn't achieve completely accurate cell-to-cell segmentation with several clusters appearing. Although this is not lethal it would be good to make this explicit in the manuscript and indicate how was this treated.

7. It would think that to most aficionados Figure1B isn't hugely convincing with regards automated binning of cellular classes. Proportionally the algorithm does a good job separating lymphocytes (TILs) and melanoma cells, but the few stromal and other cells are clearly also melanoma cells whereas the H&E on the left shows some fibroblasts/macrophage-type cells which appear to be segmented as melanoma cells. The proof is in the pudding and the algorithm is a prognostic classifier but I am not sure this example will do it justice in the community going forward.

8. The authors refer to the algorithm as open source. That is true in the sense that it was written on an open source platform, but in the methods the authors make it clear that the software will be shared upon request. To lower the threshold for prospective users, I strongly feel that this software should either be directly available on GitHub and/or be made available through QuPath. That is in line with Open Science and would also allow the community to work on issues such as the one outline in point 7.

In summary, the revised manuscript now acknowledging that these retrospective data cannot yet be used to make decisions on sparing adjuvant immunotherapy addresses our main concern with the original manuscript. This study provides a useful prognostic tool for assessment of melanoma tumour infiltrating lymphocytes with real-world patient outcome data. It will be interesting to see how it performs when tested in a prospective clinical trial.

Marnix Jansen, University College London

Reviewer #1 (Remarks to the Author):

Thank you for addressing my comments. Overall, this was/is a well articulated manuscript and response letter. I think this study is novel and will be found to be of value to the community of pathologists and oncologists as it will indeed provide an inexpensive tool to tabulate TILs using routine H&E stained sections. I recommend acceptance. One important comment in response to your comments.

1. Only CD4, CD8, and CD20 are shown in Figure 2, but CD3 was/is stated to have been tested? In any case, the point was not that it would be important to molecularly define what eTILs% is tabulating per se (I agree that would be missing the point, but only sort of). The intention was to suggest that it would be important to confirm that eTILs% is actually a measure of 'immune cells'--not that it is important what the cells actually are, provided they are indeed 'immune cells'. The contention that eTILs% is a measure of L's is a bit shaky since eTILs% only weakly correlates with conventionally tabulated metrics of immune cells (CD4, CD8 in particular). Might eTILs% correlate with the sum of CD4 and CD8 (this should be comprise most all of the T cells and many but not all of the histiocytes/dendritic cells in the sample)?

Thank you very much for taking the time to review our manuscript and thank you for your positive comments. We did not test CD3, it was mistakenly stated in the manuscript since we have tested it in other cohorts. Thank you for raising this point. Accordingly, the manuscript has been updated.

Upon your request, here you can see the correlation of eTIL% with the sum of CD4 and CD8.

As you can see, the regression has improved compared to CD4 and CD8 alone. We found a significant relationship and fair correlation between eTILs and the sum of CD4 and CD8 (Spearman $r=0.466$, $p<0.001$). A possible explanation could be immune heterogeneity in the tumor as the HE and CD4, CD8, CD20 IF stainings were not performed on serial sections. Another possible explanation is that CD4, CD8 and CD20 (immune markers) scores were calculated differently during the automated quantitative fluorescent investigation. Regarding immune markers, scores were calculated in terms of the number of cells positive for the marker of interest as a percentage of the cell population in which it was measured, while eTIL% was defined as $(\text{TILs}/\text{TILs}+\text{Tumor cells})\times 100$.

Reviewer #3 (Remarks to the Author):

This manuscript has been revised and responses provided to the reviewer's comments. We respond as follows:

Clinical context

1. The authors have now clarified that based on this retrospective study eTIL% cannot yet be used as a decision aid to spare patients from adjuvant immunotherapy. This will require testing in a prospective clinical trial, but it is accepted that the current study provides useful baseline data for such a future study.

Thank you very much for taking the time to review our manuscript and thank you for your positive comments. We agree with the reviewer, and are already making plans to test this algorithm for clinical utility in other cohorts.

2. The additional details provided on development of the algorithm and features used for neural network classification are helpful (see below).

Thanks, no response needed.

3. Whilst the authors state that eTIL% was prognostic irrespective of whether TMA or whole slide images were used, correlation between the two was not strong. This suggests there may be heterogeneity in eTIL% dependent on the region of tumour tested which could impact on the prognostic scoring. Have the authors formally tested this using feature extraction at low and higher power? Future validation should determine the optimal number of tumour regions to test for most optimal utility of the tool.

This is a very good point. Yes, the region of interest (region of tumor) has effect on the results and prognostic scoring. Although we were not attempting to validate TMA technology, we think that this might be due to the investigated tumor area on an average melanoma whole slide is 20-25-fold larger compared to a TMA spot. For clinical purposes, a TMA spot is not representative considering the whole tumor. For this reason, we tested the locked down algorithm on two independent whole slide cohorts. In the whole slide cohorts (cohort #2 and #4), the analysis was run on the entire tumor area (feature extraction at 20x zoom) following the pathologist markings (annotation), refer to Suppl Figure 1. We recommend to test the whole tumor are for clinical setting in order to avoid bias.

The reviewer also makes an interesting point related to determining the number of tumor regions (or the number of fields of view (FOV)) required to be representative. We have addressed this issue before in breast cancer (Mani et al, Breast Can Res 2016), but since the algorithm calculates so quickly it is unlikely that we would ever read a subset of the FOVs in the future for either research or in clinical setting

Computational/Histopathology

4. Both the watershed and neural networks approaches have been applied widely in order contexts to overcome segmentation and classification challenges and there is little novelty in this regard. The authors should refer to these works in the introduction-(related works) section.

The introduction has been updated according to the reviewer's comment.

5. What is the reason that the authors used the aforementioned approaches and not any other approach (Graph Cuts, RCNN, FCN)?

We wanted to use an open source, widely used and well documented platform, that could be applied by pathologist in clinical practice even in lower resourced settings. Also, achieving high quality (segmentation) results with FCN requires large scale datasets for training (tens of thousands of cells) with precisely annotated segmentation masks accurately representing the cell borders or the exact locations of all cells of the relevant types in training images, which is a tremendous manual annotation effort and prone to variation. R-CNN is well-established in computer vision for the detection of man-made objects or human body parts. However, applying these techniques for identification of cell types, including lymphocytes, may be much more challenging than to man-made objects, as histomorphological variability is high (particularly in cancer).

6. From Figure 1B, it is clear shown that the watershed algorithm didn't achieve completely accurate cell-to-cell segmentation with several clusters appearing. Although this is not lethal it would be good to make this explicit in the manuscript and indicate how was this treated.

There is no algorithm that can achieve 100% accurate cell to cell segmentation, because of tissue handling – and staining artifacts. The quality control of the cell segmentation was performed by a pathologist. This has been noted now in the manuscript.

7. It would think that to most aficionados Figure 1B isn't hugely convincing with regards automated binning of cellular classes. Proportionally the algorithm does a good job separating lymphocytes (TILs) and melanoma cells, but the few stromal and other cells are clearly also melanoma cells whereas the H&E on the left shows some fibroblasts/macrophage-type cells which appear to be segmented as melanoma cells. The proof is in the pudding and the algorithm is a prognostic classifier but I am not sure this example will do it justice in the community going forward.

We share the algorithm and all the cell segmentations setting that would allow the community to validate the utility and evaluate the impact of false detections. We agree that there may be a fair number of segmentation inaccuracies, but these apparently do not effect the prognostic value.

8. The authors refer to the algorithm as open source. That is true in the sense that it was written on an open source platform, but in the methods the authors make it clear that the software will be shared upon request. To lower the threshold for prospective users, I strongly feel that this software should either be directly available on GitHub and/or be made available through QuPath. That is in line with Open Science and would also allow the community to work on issues such as the one outline in point 7.

Agree, the algorithm has been deposited on GitHub: <https://github.com/acsbal/Automated-TIL-scoring-QuPath-Classifer-for-Melanoma>

The detailed documentation how to use the algorithm is in progress and will be added in advance of publication to facilitate community access to this test.

In summary, the revised manuscript now acknowledging that these retrospective data cannot yet be used to make decisions on sparing adjuvant immunotherapy addresses our main concern with the original manuscript. This study provides a useful prognostic tool for assessment of melanoma tumour infiltrating lymphocytes with real-world patient outcome data. It will be interesting to see how it performs when tested in a prospective clinical trial.

Once again, we would like to thank the editor and reviewers for their comments and suggestions. We hope that we have been able to satisfy these revisions.

Sincerely,

Balasz Acs and David L. Rimm, MD-PhD

REVIEWERS' COMMENTS:

Reviewer #3 (Remarks to the Author):

The authors have satisfactorily addressed remaining concerns. This is an important piece of work that now merits publication.

Reviewer #3 (Remarks to the Author): The authors have satisfactorily addressed remaining concerns. This is an important piece of work that now merits publication. Thank you very much for reviewing our manuscript and for your positive comments.